# *Rhus Coriaria* L. Extract: Antioxidant Effect and Modulation of Bioenergetic Capacity in Fibroblasts from Parkinson’s Disease Patients and THP-1 Macrophages

**DOI:** 10.3390/ijms232112774

**Published:** 2022-10-23

**Authors:** Camilla Isgrò, Ludovica Spagnuolo, Elisa Pannucci, Luigi Mondello, Luca Santi, Laura Dugo, Anna Maria Sardanelli

**Affiliations:** 1Department of Translational Biomedicine Neuroscience ‘DiBraiN’, University of Bari “Aldo Moro”, Piazza G. Cesare 11, 70124 Bari, Italy; 2Department of Medicine, University Campus Bio-Medico of Rome, Via Alvaro del Portillo 21, 00128 Rome, Italy; 3Department of Science and Technology for Humans and the Environment, University Campus Bio-Medico of Rome, Via Alvaro del Portillo 21, 00128 Roma, Italy; 4Department of Chemical, Biological, Pharmaceutical and Environmental Sciences, University of Messina, 98168 Messina, Italy; 5Chromaleont s.r.l., c/o Department of Chemical, Biological, Pharmaceutical and Environmental Sciences, University of Messina, 98168 Messina, Italy; 6Department of Agriculture and Forest Sciences, University of Tuscia, Via San Camillo de Lellis snc, 01100 Viterbo, Italy

**Keywords:** human fibroblasts, early-onset Parkinson’s disease, PARK2, parkin, *Rhus coriaria* L., antioxidant activity, mitochondria

## Abstract

Sumac, *Rhus coriaria* L., is a Mediterranean plant showing several useful properties, such as antioxidant and neuroprotective effects. Currently, there is no evidence about its possible neuroprotective action in Parkinson’s disease (PD). We hypothesized that sumac could modulate mitochondrial functionality in fibroblasts of familial early-onset PD patients showing PARK2 mutations. Sumac extract volatile profile, polyphenolic content and antioxidant activity have been previously characterized. We evaluated ROS and ATP levels on sumac-treated patients’ and healthy control fibroblasts. In PD fibroblasts, all treatments were effective in reducing H_2_O_2_ levels, while patients’ ATP content was modulated differently, probably due to the varying mutations in the PARK2 gene found in individual patients which are also involved in different mitochondrial phenotypes. We also investigated the effect of sumac extract on THP-1-differentiated macrophages, which show different embryogenic origin with respect to fibroblasts. In THP-1 macrophages, sumac treatment determined a reduction in H_2_O_2_ levels and an increase in the mitochondrial ATP content in M1, assuming that sumac could polarize the M1 to M2 phenotype, as demonstrated with other food-derived compounds rich in polyphenols. In conclusion, *Rhus coriaria* L. extracts could represent a potential nutraceutical approach to PD.

## 1. Introduction

Parkinson’s disease (PD) is a slow progressive neurodegenerative disorder characterized by a loss of dopaminergic neurons and results in motor system dysfunctions such as bradykinesia, tremors, muscular rigidity and postural instability. Most cases of PD are idiopathic, while a small percentage results from specific genetic mutations [1,2]. Although PD is considered an age-related disorder, it also affects younger individuals. Indeed, about 3.6% of all patients with PD show a disease onset before the age of 45, which is defined as early-onset PD (EOPD) [3]. EOPD is genetically heterogeneous, and some major EOPD genes have been described, with both autosomal recessive (PARK2, PINK1 and PARK7 genes) and dominant (LRRK2 gene) transmission. Among these genes, mutations in PARK2 and PINK1, coding for E3 ubiquitin–protein ligase Parkin and for a serine/threonine protein kinase, respectively, are the most common causes of EOPD [4,5]. The Parkin protein protects neurons against α-synuclein toxicity and oxidative stress [6]. Moreover, it can ubiquitinate mitochondrial proteins involved in mitochondrial fusion and activates mitophagy to scavenge depolarized mitochondria [7].

Several findings highlighted the importance of mitochondria in the pathophysiology of EOPD [8]. Indeed, some of the gene mutations associated with EOPD affect mitochondrial dynamics and function, inducing parkinsonism [5,9,10]. In particular, PARK2 has a pivotal role in mitochondrial homeostasis: parkin may promote mitochondrial division [11]; in association with PINK1, it acts in mitochondrial fusion and transport; through a mitophagic process, it removes damaged mitochondria [5]. Thus, the deficiency of PARK2 or PINK1 gene products bring several alterations in mitochondrial functions, resulting in decreased energy production and induction of stress-induced apoptosis [12]. Reactive oxygen species (ROS) represent a normal outcome of metabolic processes and, in moderate amounts, play important roles in cellular defense. Mitochondria represent one of the principal sources of ROS, and ROS levels within cells are kept in check through an endogenous antioxidant enzyme defense system [13]. However, excessive ROS generation can deplete this system, causing oxidative stress, constantly associated with the pathogenesis of PD [12,14].

It has been demonstrated that dopaminergic neurons obtained from induced pluripotent stem cells (iPSCs) carrying Parkin mutations showed an increase in ROS production [15,16]. On the other hand, oxidative stress down-regulates E3 ligase activity of the Parkin protein [17]. These findings suggest that Parkin can indirectly modulate oxidative stress and mitochondrial quality processes by mitophagy, while oxidative stress impairs its activity [18].

Skin fibroblasts are recognized as a primary human cell model useful as a peripheral marker of PD [19], as they are an easily accessible source of proliferating cells that share the same genetic complexity as neurons in order to overcome the limitations of the conventional research approach in the field of neurodegenerative diseases, where post mortem examination of brain tissue still prevails [20,21,22]. Indeed, it is documented that Parkin is expressed in human skin fibroblasts [23].

Therefore, the study of mitochondrial function may be crucial for investigating PD pathogenesis and for developing prevention strategies and more effective therapies even in the initial stages of PD. Considerable attention is now focused on the potential therapeutic role of antioxidants in neurodegenerative diseases such as PD [24].

Several studies have shown that dietary substances, and in particular plant-derived secondary metabolites, can have a protective action against oxidative stress and neuroinflammation, which are hallmarks of neurodegenerative diseases [25]. The ability to cross the blood–brain barrier plays an important role in the neuroprotective action of natural products of bioactive compounds [25,26]. Polyphenols, a wide group of phytochemicals present in plants, fruits and vegetables, have gained growing attention due to their high capacity to prevent and reduce the harmful effects of oxidative stress. For example, resveratrol has significant neuroprotective activity both in vitro and in vivo [27,28,29]. Curcumin can induce neuroprotective effects through the control of pathogenetic oxidative and inflammatory mechanisms in both in vitro and in vivo models of PD [30]. Green tea extracts and the isolated (–)-epigallocatechin-3-gallate polyphenol have neuroprotective effects, being able to decrease the dopaminergic neuron loss in the substantia nigra and the oxidative damage in PD animal models [31]. Taken together, this evidence suggests that polyphenols act as neuroprotective agents, activating several molecular pathways [32]. The daily consumption of dietary polyphenols, in fact, inhibits various secondary sources of ROS and proinflammatory cytokines, reducing the risk of developing neurodegenerative diseases, configuring a useful and promising approach for prevention and treatment.

*Rhus coriaria* L., commonly known as sumac, is a Mediterranean plant traditionally used as a spice and flavoring agent. It belongs to the Anacardiaceae family, and it is diffused in temperate and tropical regions worldwide, frequently growing in areas with marginal agricultural capacity. Since ancient times, various components of the sumac plant have been used for many applications. For example, tannins extracted from young stems and from leaves were used for tanning hides in animal skin preparation and within the past centuries the foremost in-depth plantations were established for this purpose. Moreover, sumac bark and fruit preparations are extensively employed in standard medication to obtain natural remedies against various ailments of ophthalmological, infectious and gastro-intestinal origin [33]. *Rhus coriaria* L. extracts’ volatile profile, polyphenolic content and antioxidant activity have been characterized recently by our research group [33]. From this analysis, 263 volatile compounds and 83 polyphenolic compounds were positively identified in the investigated samples; most of the latter were represented by gallic acid (GA) and its derivates. *Rhus coriaria* plants have a long history in traditional medicine especially in the countries of the Middle East and South Asia. Current studies suggest that this plant and its extracts are very safe, making them attractive for medicinal use or drug discovery approaches. In fact, sumac shows a wide range of pharmacological and biological activities, including antioxidant, antimicrobial, antidiabetic, cardioprotective, antidyslipidemic, neuroprotective and anticancer effects [34,35]. However, there is very poor evidence about the neuroprotective effects of sumac extract [36,37,38,39], and none about a possible effect in PD. Among these, the antioxidant properties are very promising for the prevention and/or the treatment of neurodegenerative disorders such as PD. Therefore, we hypothesized that the antioxidant compounds naturally found in the sumac extract could prevent, delay or alleviate the clinical symptoms of PD by counteracting some of the main pathophysiological mechanisms involved in the development of the disease, such as oxidative stress and neuroinflammation. In this context, the aim of the work was to test if a phenolic *Rhus coriaria* L. extract (sumac extract, SE) could be effective in the modulation of ROS production and mitochondrial functionality (through ATP oxidative synthesis) in primary fibroblasts of patients affected by EOPD (PARK2 mutation) compared with the healthy control.

In order to confirm the role of SE as an antioxidant, we also investigated its effect on THP-1 differentiated macrophages, which show different embryogenic origins with respect to fibroblasts and which represent cell models for the study of ROS modulation [40]. These analyses had an exploratory aim and could eventually lead to further studies on primary macrophages obtained from PD patients.

## 2. Results

### 2.1. Antioxidant Activity of Phenolic Sumac Extract

Results previously obtained [33] identified GA and its derivatives as the main components of the SE; thus, the antioxidant activity of SE and GA was determined as indicated in Materials and Methods.

The 2,2-Diphenyl-2-picrylhydrazyl Hydrate (DPPH) and 2,2′-azinobis (3-ethylbenzothiazoline-6-sulfonic acid) (ABTS) radical scavenging potential of *Rhus coriaria* extract and GA is reported in Table 1, and results are expressed as Trolox Equivalents (mmol TE/g extract) and as IC50 (mg/mL), defined as the amount of antioxidant necessary to decrease the initial radical concentration by 50%. As a result, SE showed a remarkable antioxidant activity; moreover, GA exhibited a strong antioxidant capacity, indicating that GA played a key role as active component in the overall antioxidant activity of SE.

### 2.2. Effect of Phenolic Sumac Extract on Cell Viability

To test the effect of SE on cell viability, primary skin fibroblasts from PD patients (indicated as Pt1, Pt2 and Pt3) [41,42,43] listed in Table 2 were incubated for 24 and 48 h in the presence of increasing concentrations of SE (3.125, 6.25, 12.5 μg/mL gallic acid equivalents, GAE), and data were compared with normal human dermal fibroblasts (NHDFs) as control. Total phenolic content was determined with the Folin–Ciocalteu method from a GA standard curve, and the results were expressed as µg of GAE per ml, according to Arena et al., 2022 [33]. The residue cell viability was measured through MTT assay, and the results were expressed as % vitality against the untreated control.

The treatment with increasing concentrations of SE (expressed as μg/mL GAE) affected the cell viability of the analyzed samples differently. In fact, Figure 1 shows that increasing concentrations of SE did not significantly affect cell viability in NHDFs (Figure 1a) and Pt3 (Figure 1d) with respect to the untreated controls. On the contrary, in Pt1 (Figure 1b) and Pt2 (Figure 1c), a great decrease in the residue cell viability was observed at both 6.25 and 12.5 μg/mL GAE, showing a significant dose- and time-dependent manner. Moreover, the difference remained significant also comparing the three different concentrations of SE among themselves. Considering these results, we selected for further treatment of PD fibroblasts the 3.125 and 6.25 μg/mL GAE concentration values and we chose 24 h as the end point for treatment, because at this time point cell viability was best maintained at all the tested concentrations.

The effect of phenolic SE on cell viability was also tested in THP-1 macrophages at increasing concentrations (3.125, 6.25, 12.5, 25, 50 μg/mL GAE). As shown in Figure 2, SE at up to 50 μg/mL GAE concentration failed to display toxicity, and after 48 h of treatment an increase is observable in cell viability at higher concentrations (25, 50 μg/mL GAE) in M0 phenotype (Figure 2a). However, SE on the M1/M2 phenotype did not show negative effects on cell viability in the time points considered, and no differences between all the concentrations tested were observed (Figure 2b,c).

### 2.3. Effect of Phenolic Sumac Extract Treatment on H_2_O_2_ Cellular Production

ROS production was determined with the H_2_DCF-DA assay that measures the cellular H_2_O_2_ levels spectrophotometrically. Cells were seeded in a 96-well black plate and incubated for 24 h with 3.125 and 6.25 μg/mL GAE. Results are expressed as % compared to the untreated controls. As shown in Figure 3, PD patients showed higher H_2_O_2_ levels compared to untreated healthy control NHDFs, indicating that in these patients oxidative stress has occurred.

Looking at the effect of SE treatment on NHDF (Figure 4a) and PD patients (Figure 4b–d), we can observe that the H_2_O_2_ levels measured after the treatment with 3.125 and 6.25 μg/mL GAE resulted as significantly decreased comparing Pt1, Pt2 (*p* value < 0.001) and Pt3 to the NHDFs (*p* value < 0.01), in a dose-dependent manner.

Figure 5 shows the differential effects of SE at 3.125, 12.5 and 50 μg/mL GAE on ROS production in macrophage phenotypes: no change is observed in M0 macrophages (Figure 5a) after treatment with respect to the untreated control. In the M1 phenotype (Figure 5b) we detected a dose-dependent decrease, while in the M2 phenotype only 50 µg/mL GAE resulted in a significant decrease in H2O2 production compared to the M2 untreated control (Figure 5c).

### 2.4. Effect of Phenolic Sumac Extract on Bioenergetic Capacity

After the determination of the H_2_O_2_ levels following the treatment with SE, we decided to investigate if it could influence ATP production in both control and PD patients’ cells. We evaluated the effect of SE not only on the total ATP content, but also considered the rotenone (inhibitor of mitochondrial Complex I, CI) and antimycin A (inhibitor of mitochondrial Complex III, CII-III) sensible quotes to better understand if SE may have an effect in modulating glycolytic or oxidative pathways. First, we determined if the ATP content was different in untreated healthy control and patients’ fibroblasts. In Figure 6, the ATP content registered in the untreated cells (in the absence of SE) shows a great difference between the NHDF and all the PD patients analyzed. Both Pt1 and Pt2 show a significant decrease in the total ATP levels (*p* value < 0.001). Even Pt3 presents a significant reduction in the total ATP levels, although less noticeable than the other two patients (*p* value < 0.01). These differences remain remarkable in Pt1 and Pt2 even when the cells were treated with rotenone and antimycin A (*p* value < 0.001), namely glycolytic CI inhibition and glycolytic CII-III inhibition, respectively. Finally, mitochondrial ATP levels (indicated as mitochondrial CI activity and mitochondrial CII-III activity) revealed to be strongly reduced in all the patients compared to NHDF (*p* value < 0.001), indicating an impairment of mitochondrial functionality and energetic capacity in these patients.

As shown in Figure 7a, SE did not have any effects on the ATP production in NHDF. Only a slight decreasing trend is noticeable compared to the untreated controls. In fact, the minor differences that we can appreciate in the graph are indeed not statistically significant (*p* value ns).

On the contrary, the same treatment had an influence on the ATP production in Pt1. In fact, Figure 7b shows how even if the treatment with 3.125 μg/mL GAE of SE determined only a slight increase in the total ATP levels (*p* value < 0.05), it caused a significant increase in the antimycin A-treated ATP quote (*p* value < 0.01), associated with a slight decrease in the mitochondrial ATP quote derived by CII-III activity (*p* value < 0.05). Looking at the influence of our treatment on the ATP content in Pt2 cells, Figure 7c shows different results. In fact, even if the total ATP content was not altered following the administration of the SE with respect to the untreated control, the CII-III inhibition after 3.125 μg/mL GAE treatment resulted in a significant reduction in the glycolytic antimycin A sensible quote compared to the one registered in the untreated control (*p* value < 0.01). A rise in mitochondrial ATP levels thanks to the activity of CII-III complexes was also detected (*p* value < 0.01). Figure 7d shows the results obtained following the treatment with SE in Pt3 cells, both in the presence and in the absence of rotenone and antimycin A. Total ATP levels were significantly affected by 3.125 μg/mL GAE treatment, resulting in a decrease in the total ATP content (*p* value < 0.05). This result is associated with a significant decrease also in the antimycin A treated sample compared to the untreated control (*p* value < 0.05) and in an increase in the relative mitochondrial ATP quote (*p* value < 0.05).

The macrophage ATP metabolic rate was also evaluated in the differentiated THP-1-derived M0, M1 and M2 macrophages. As shown in Figure 8a, ATP levels were reduced in M0 and M2 phenotypes in the presence of both mitochondrial inhibitors (** *p* < 0.01) but not in M1 macrophages (*p* = ns) that exhibit glycolytic metabolism. Indeed, the mitochondrial ATP quote of M1 is much lower than that of M2 macrophages. Following SE treatment, we can observe that in glycolytic conditions there is no difference after exposure to 3.125 or 50 µg/mL GAE concentrations both in M0 (Figure 8b) and M2 (Figure 8d) macrophages in comparison to untreated cells. Therefore, SE did not influence ATP levels in these two phenotypes. On the contrary, in M1 macrophages, SE treatment determines a reduction in ATP levels in the presence of rotenone and antimycin A in favor of an increase in the mitochondrial ATP quote (Figure 8c).

## 3. Discussion

Parkinson’s disease (PD) is the second most common neurodegenerative disease after Alzheimer’s disease. The three predominant pathways that can trigger the neurodegenerative process are: (a) accumulation of aggregated and misfolded proteins; (b) impairment of the ubiquitin–protein pathway (UPS) and the autophagy pathway; (c) mitochondrial dysfunction [44]. Although most cases of PD are sporadic, numerous familial forms of Parkinsonism have been described in which there is a Mendelian mode of transmission of the disease [2]. In particular, the PARK2 gene encodes for the Parkin protein, and Parkin mutations account for the most known causes of EOPD [4]. The Parkin protein acts as an E3 ubiquitin ligase that is recruited to damaged mitochondria to ubiquitylate outer membrane proteins to promote clearance through mitophagy and autophagic pathways [45]. Several studies reported a connection between the mutation of Parkin and mitochondrial processes, such as mitophagy, ROS production, decreased ATP levels and loss of mitochondrial membrane potential (Δψm) [5,9,10,11].

Since phytochemicals are shown to have protective action against oxidative stress and neuroinflammation [46], we hypothesized that SE could be protective against the oxidative stress typically associated with neurodegenerative diseases. Our data revealed that in PD patients carrying different PARK2 mutations, the H_2_O_2_ levels were significantly higher with respect to the healthy control (NHDF), indicating that an oxidative stress status has occurred in these patients (Figure 3), even if an increasing trend in ROS levels is present in PD patients (*p* value ns). Furthermore, interesting results were obtained from the ATP assay, comparing untreated patients vs. NHDFs. In fact, as shown in Figure 6, PD patients present a deficit in ATP production with respect to the healthy control.

Total, glycolytic and mitochondrial ATP levels were significantly reduced in PD patients with respect to NHDF. Moreover, PD patients’ glycolytic levels were not significantly different from the relative total levels, indicating that in these patients an impairment of mitochondrial functionality has occurred, and glycolysis is trying to compensate the mitochondrial energy failure.

Firstly, we tested the possible cytotoxic effect of SE (expressed as GAE, indicating the total phenolic content) on PD fibroblast cells (after 24 and 48 h of treatment), to assess the best concentrations and timing to perform other experiments. Results obtained from the MTT test showed SE to be safe at 3.125 and 6.25 μg/mL GAE doses, while 12.5 μg/mL GAE resulted as highly toxic in Pt1 and Pt2. Based on these results, we decided to proceed with the dosage of total H_2_O_2_ levels excluding the 12.5 μg/mL GAE concentration from the analysis, treating samples for 24 h. SE did not show any effect in our healthy control. On the contrary, in Pt1, the treatment with both 3.125 and 6.25 μg/mL GAE resulted in a great reduction in the total H_2_O_2_ levels with respect to the untreated control. Pt2 ROS levels were significantly decreased following the treatment with the highest dose of SE. Finally, in Pt3, only the highest dose tested revealed to be beneficial in reducing ROS.

All these data are according to the state of the art. In fact, it is well-defined that flavonoids can act as antioxidants by themselves, because they have ROS scavenging activity, which is derived from their chemical feature [47].

In general, the comparison between patients and healthy control shows that, as already explained, the ATP content is significantly lower in patients’ cells than in NHDFs, even in the absence of any treatment. These results agree with the current literature [5] and with previous data obtained by our research group [48] where the total cellular ATP content, measured under basal conditions, was noticeably decreased in the patients with respect to the control (with a maximal decrease observed in Pt2). Interestingly, also here the ATP levels obtained under strict glycolytic conditions, i.e., in the presence of rotenone, are quite like that observed in non-glycolytic conditions. All these findings support the notion that the effective ATP production by the Oxidative Phosphorylation System (OXPHOS) is compensated by glycolytic supply in PD patients. This could be due to the fact that in these patients (but not in Pt1) ATP synthase may act as an ATPase, hydrolyzing and not synthetizing ATP ([48] and manuscript in preparation).

In this work, no noticeable changes were found in ATP levels (both total and glycolytic) after the treatments compared to the untreated control in NHDFs. Conversely, the ATP levels were greatly influenced in the PD patients’ cells after receiving SE treatments. In fact, in Pt1 we can appreciate a slight increase in the total ATP levels, accompanied by a significant increase in the glycolytic (antimycin A sensible) quote and a decrease in the mitochondrial one following the administration of 3.125 μg/mL GAE of SE. In Pt2, total ATP content remained stable after the treatment but, interestingly, 3.125 μg/mL GAE of SE reduced the glycolytic ATP content measured after the antimycin A treatment, suggesting that this phenolic extract may have a positive effect on the oxidative pathway in this patient, as confirmed by mitochondrial (CII-III activity) levels shown in Figure 7c. Finally, in Pt3 the treatment caused a decrease in the basal and antimycin A treated quote of ATP content accompanied by an increase in mitochondrial ATP levels, with respect to the untreated control. Based on the different treatments, since SE did not have a strong effect on the total ATP values in parkinsonian patients, the changes in the glycolytic quotes are not sufficient to make up for the gap that existed between their ATP levels and those of the healthy control. Furthermore, it must be remembered that, although several studies have already demonstrated the antioxidant effect of the SE, this is the first work that investigated its possible role in Parkinson’s disease, particularly referring to early-onset parkinsonism.

Finally, given the heterogeneity of these results, we can conclude that this extract influences ROS levels and ATP production, but that this effect differs according to the type of mutation carried by these patients. Previous data obtained from our research group found that the basal endogenous respiration rate measured in these patients was notably lower with respect to the control. This decrease was maintained in all patients except for Pt1, even after the addition of 2,4-dinitrophenol (DNP), a molecule that acts as uncoupler of the OXPHOS system, and even after the addition of the exogenous substrates supplying electrons to all the OXPHOS complexes [48].

The effect of SE was also evaluated in the THP-1 monocyte-derived macrophage cell line to investigate the antioxidant role of this natural compound. Macrophages play a pivotal role in immunological innate response in vivo and also contribute to tissue repair processes [49].

Moreover, macrophages have tissue-specific functions and can rapidly change their activation states in response to different stimuli. Indeed, macrophages can reprogram their method of generating ATP for energy depending on their polarization state: the anti-inflammatory M2 phenotype is more dependent on oxidative metabolism, while the pro-inflammatory M1 phenotype is mainly based on a glycolytic metabolism [50,51].

Since it has been previously demonstrated that phenolic-rich compounds can favor macrophage polarization in the M2 anti-inflammatory phenotype, attenuating the M1 activation response, in our study we evaluated if SE could exert the same effect in THP-1-derived macrophages [52,53,54]. However, it is important to emphasize that our present aim was not to demonstrate that SE treatment was able to determine a change in the polarization state of macrophages, but only possible changes in ROS production and bioenergetic capacity in these cells, so M1/M2 markers were not determined in this study.

We demonstrated that SE treatment decreases ROS production in M1 in a concentration-dependent manner, while in the M2 phenotype only the highest extract concentration reduced ROS, compared to M2 untreated cells and the other concentrations used. In M0 no changes were observed, as expected.

We also evaluated if SE could influence ATP levels in macrophages. In the absence of the polyphenolic compounds, in the M1 phenotype, glycolytic and total ATP levels were similar, while in the M2 phenotype the glycolytic quote was significantly reduced with respect to total levels. In addition, the ATP mitochondrial quote in M1 was reduced compared to the M2 phenotype, confirming our knowledge about the different metabolism typical of the two activation states. Considering basal levels of ATP, we observed (Figure 8a) that M1 shows very low basal levels of ATP compared to other macrophage phenotypes, probably due to an ATP release from cells after their activation with LPS, as demonstrated by Sakaki et al., 2013 [55].

On the contrary, following SE treatment in both M0 and M2 macrophages there were no changes in ATP levels overall. In SE-treated M1 we observed a reduction in total levels of ATP that is associated also to a strong reduction in the glycolytic ATP quote, suggesting that this compound could favor a metabolic reprogramming of M1 from a glycolytic metabolism to a more oxidative one, eventually leading to a reduction in systemic inflammation. These promising results need to be confirmed with further in vitro studies on PD patients’ derived macrophages to definitely assess if SE could be effective in reducing oxidative stress favoring the activation of a macrophage anti-inflammatory phenotype.

## 4. Materials and Methods

### 4.1. Samples, Cell Growth and Differentiation

The experimental cell model that was used in this study was primary cultures of control human fibroblasts (NHDF) and primary cultures of fibroblasts isolated by skin biopsy from PARK2-mutant patients affected with early onset form of PD (cell passage range 5–8) and THP-1 cell line. Three Italian patients showing an early onset form of PD were selected for this study (Table 2). Diagnosis of PD was made according with the UK Brain Bank criteria evaluating the Unified Parkinson’s Disease Rating Scale (UPDRS) and Hoehn–Yahr Scale. Primary skin fibroblasts were obtained by explants from skin punch biopsy. Skin biopsies were obtained after informed consent. Age-matched adult normal human dermal fibroblasts (NHDF, Lonza, Walkersville Inc. Walkersville, MD, USA) were used as healthy control.

Primary fibroblasts were grown in Dulbecco’s modified Eagle’s medium High Glucose DMEM (5 mM glucose), supplemented with 10% heat-inactivated fetal bovine serum FBS, 1% L-glutamine, 1% penicillin–streptomycin (Euroclone S.p.A.,Pero, Italy), in a 37 °C, 5% CO_2_ (pH 7.2–7.4) incubator. THP-1 cell line (ATCC: TIB-202) was cultured in RPMI 1640 medium (Corning, NY, USA) supplemented with 100 U/mL penicillin, 100 μg/mL streptomycin (Corning, NY, USA), 10 mM Hepes (Dominique Dutscher, Bernolsheim, Francei), 2 mM glutamine (Corning, NY, USA) and 10% (*v*/*v*) heat-inactivated fetal bovine serum FBS (Corning, NY, USA) at 37 °C in humidified atmosphere containing 5% (*v*/*v*) CO_2_. THP-1 cells can be differentiated into macrophage-like cells M0, M1-polarized or M2-polarized cells according to the method described by Dugo et al. 2017 [54].

### 4.2. Cytotoxic Effect

Fibroblasts of both patients and control were seeded in a clear 96-multiwell plate at concentration of 12 × 10^3^ cells/well and allowed to adhere overnight. Then, the cells were incubated in the absence (untreated control) and in the presence of SE for 24 and 48 h as 3.125–6.25–12.5 μg/mL GAE for the cytotoxic analysis on PD patients’ cells. Macrophages differentiated (50 × 10^3^ cells per well in 96-well cell culture plates) were exposed to different concentrations of SE as 3.12–6.25–12.5–25–50 μg/mL GAE at the same time for fibroblasts. Then, cell culture medium was discarded, and each well was washed with 200 μL PBS (Euroclone S.p.A., Pero, Italy). The MTT solution (0.5 mg/mL in serum-free medium, 100 μL Sigma-Aldrich, Milan, Italy) was added to cells in each well, and the plate was incubated at 37 °C, 5% CO_2_ for 3 h. Then, MTT solution was removed, and dimethyl sulfoxide (DMSO, Sigma-Aldrich, Milan, Italy) (150 μL/well) was added to each well for dissolving the formazan crystals. Optical density (OD) was measured at λ 570 nm using a multifunctional microplate reader (Infinite 200 Pro, TECAN, Italy). Viability was calculated as the ratio of the mean of OD obtained for each condition to that of the untreated control condition and expressed as % of control.

### 4.3. ROS Production

Cells were grown in 96-multiwell black plates at cell densities of 12 × 10^3^ cells/well for fibroblasts and 50 × 10^3^ cells/well for macrophages. Once attached, cells were treated for 24 h with 3.125 and 6.25 μg/mL and 3.125–12.5–50 μg/mL of GAE, for fibroblasts and macrophages, respectively. At the end of the treatment period, H_2_DCF-DA probe (Merck Millipore, Burlington, MA, USA) 20 μM was added to each well and was incubated at 37 °C, 5% CO_2_ in the dark for 40 min. In the presence of ROS, DCF is oxidized to highly fluorescent DCF, which can be detected and quantified by fluorimetric assay, which was carried out on the multifunctional multiplate reader (Infinite 200 Pro, TECAN, Italy) at λ Ex/Em = 485/535 nm. ROS (H_2_O_2_) levels detected were expressed as % of H_2_O_2_ production with respect to the healthy and/or untreated controls.

### 4.4. ATP Assay

Cellular ATP levels were measured by a luminometer (Infinite 200 Pro, TECAN, Italy) using the ATPlite kit (Perkin Elmer, Waltham, MA, USA) based on a luciferin–luciferase reaction system, according to the manufacturer’s instructions as indicated in [54]. Briefly, NHDF and patients’ fibroblasts, incubated for 24 h in the presence and in the absence of GAE, were collected by trypsinization, and then the experiment proceeded in a 96-well black plate (12 × 10^3^ cells/well). To successively perform the protein quantification using the Bradford method, 10 µL of each condition was stored [56].

Macrophages were seeded and eventually differentiated directly into a 96-well black plate at 6 × 10^5^ cells and then treated with GA equivalent for 24 h.

For the evaluation of ATP content under strict glycolytic conditions, fibroblasts were incubated for 5 h at 37 °C both in the presence of rotenone (1 μM) that of antimycin A (1 μM), while macrophages were treated with the same inhibitors combined at a final concentration of 10 μM. An ATP calibration curve was made using known concentration ATP solutions to calculate the µM concentrations of the samples’ ATP. The results were normalized for mg of total proteins for fibroblast cell lines and for 10^6^ cells for macrophages.

### 4.5. Antioxidant Activity

The antioxidant activity of SE and GA was determined by 2,2-Diphenyl-2-picrylhydrazyl Hydrate (DPPH) and 2,2′-azinobis (3-ethylbenzothiazoline-6-sulfonic acid) (ABTS) Radical Scavenging Assays. For both ABTS and DPPH assays, samples were diluted appropriately in aqueous phosphate buffer solution (PBS, 5 mM, pH = 7,4).

DPPH assay was performed as described by Bobo-García et al. [57]. Briefly, 20 μL of extract, GA (Merck KGaA, Darmstadt, Germany) or Trolox (Merck KGaA, Darmstadt, Germany), was added to 180 μL of DPPH (Merck KGaA, Darmstadt, Germany) solution (150 μmol L^−1^) in methanol–water (80:20, *v*/*v*) and shaken for 60 s in a 96-well microplate. After 40 min in the dark at room temperature, the absorbance was measured at 515 nm a multifunctional microplate reader (Infinite 200 Pro, TECAN, Italy).

The free radical scavenging capacity of SE and GA was also studied using the ABTS radical cation decolorization assay. The ABTS assay was performed as described by Re et al. [58] with some modifications. Briefly, the ABTS radical cation (ABTS^•+^) was produced by the reaction of ABTS (Merck KGaA, Darmstadt, Germany) stock solution (7 mM) with potassium persulfate (Merck KGaA, Darmstadt, Germany) (2.45 mM) and allowing the mixture to stand in dark at room temperature for 16 h before the use. The ABTS^•+^ solution was diluted in aqueous phosphate buffer (5 mM, pH = 7,4) solution to give an absorbance of 0.7 ± 0.05 at 734 nm. Then, 10 μL of SE, GA or Trolox acid were mixed with 190 μL of ABTS^•+^ diluted solution in a 96-well plate and the absorbance was recorded at 734 nm for 90 s by a multifunctional microplate reader (Infinite 200 Pro, TECAN, Italy).

For both the assays the radical scavenging activity % (RSA%) was calculated using Equation (1):(1)RSA%=1−Asample−AblankAcontrol−Ablank×100
where *A_sample_* is the absorbance of extract or GA, *A_blank_* is the absorbance of the solvent (methanol-water 80:20, *v*/*v*) and *A_control_* is the absorbance of the solvent instead the sample. The analysis was performed in triplicate. The IC50 value, referred to the concentration providing 50% of radical inhibition, was calculated from the plotted RSA% graph against the concentrations of the samples; moreover, results were expressed as mmol of Trolox Equivalents (TE)/g of extract, obtained from a Trolox standard curve (100–700 μM). Analysis was performed in triplicate.

### 4.6. Statistical Analysis

Data obtained from in vitro analyses on cells were analyzed using GraphPad Prism version 5.01. One-way ANOVA test was used to perform the analysis of variance, followed by the Tukey’s Multiple Comparison Test to compare the difference between each pair of means with appropriate adjustment for the multiple testing. The statistical analyses of the antioxidant activity of SE were performed using a *t*-test statistical procedure on Microsoft Excel (Version 2206). All the results were considered significant for *p* value < 0.05 and CI 95%.

## 5. Conclusions

Our data suggest that SE represents a promising natural compound that could be effective in the modulation of PD progression and in the attenuation of the systemic inflammatory response, thanks to its antioxidant effect and its capacity to modulate ATP production by both glycolytic and oxidative phosphorylation processes. Of course, further investigations are needed. In the future, it may be interesting to evaluate in both patients’ fibroblasts and macrophages how the single complexes of the respiratory chain act in response to the treatment with SE, expanding our research also to patients affected by idiopathic forms of PD. It may be interesting also to evaluate SE’s effect on other cellular processes in which mitochondria are involved, such as mitophagy and mitochondrial dynamics.

## Figures and Tables

**Figure 1 ijms-23-12774-f001:**
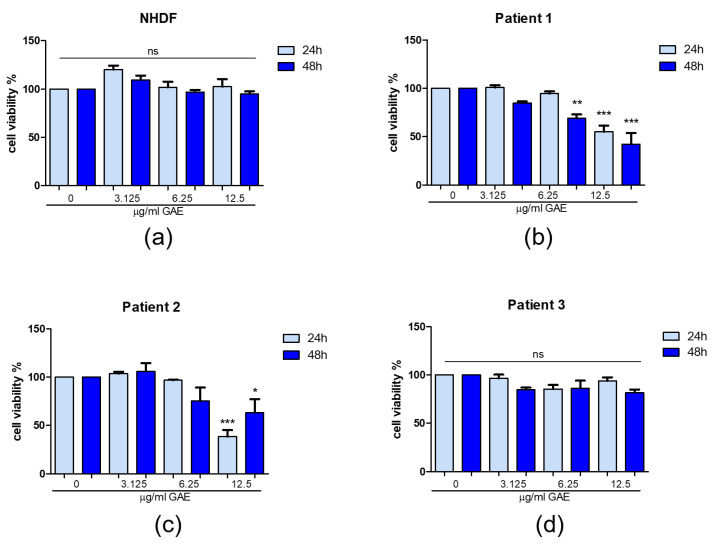
Cell viability after phenolic sumac extract treatment in NHDF (**a**), patient 1 (**b**), patient 2 (**c**), patient 3 (**d**). MTT assay was performed at 24 h and 48 h post-treatment. Cell viability was expressed as % of control (0 μg/mL GAE). All data represent the means of at least 3 replicates ± standard deviation. The analysis of variance was carried out using the One-way ANOVA test followed by the Tukey’s Multiple Comparison Test; * *p* < 0.05, ** *p* < 0.01, *** *p* < 0.001 were considered significant (untreated control vs. treated samples). NHDF: normal human dermal fibroblast; GAE: gallic acid equivalents.

**Figure 2 ijms-23-12774-f002:**
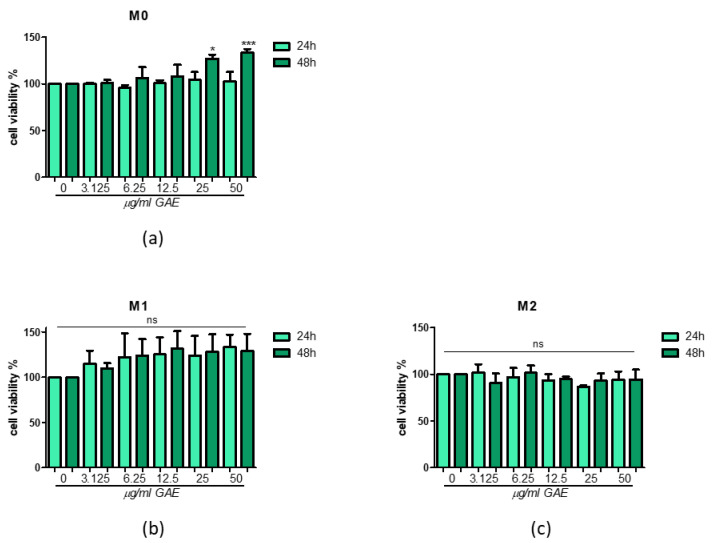
Cell viability after phenolic sumac extract treatment in M0 (**a**), M1 (**b**) and M2 (**c**). MTT assay was performed at 24 h and 48 h post-treatment. Cell viability was expressed as % of control (0 μg/mL GAE). All data represent the means of at least 3 replicates ± standard deviation. The analysis of variance was carried out using the One-way ANOVA test followed by the Tukey’s Multiple Comparison Test; * *p* < 0.05, *** *p* < 0.001 were considered significant (untreated control vs. treated samples). GAE: gallic acid equivalents.

**Figure 3 ijms-23-12774-f003:**
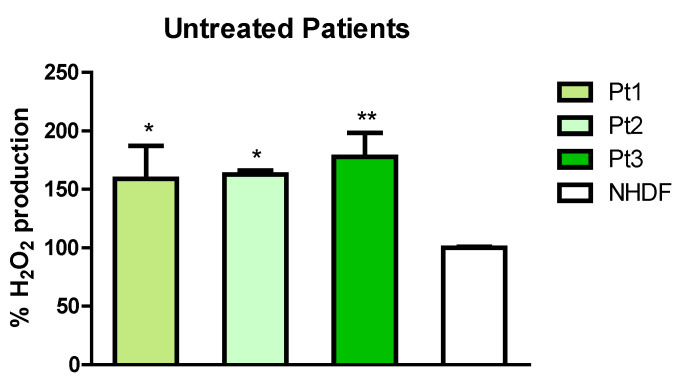
ROS production in untreated patients and NHDF. Production of intracellular ROS was expressed as % H2O2 production. All data represent the means of at least 3 replicates ± standard deviation. The analysis of variance was carried out using the One-way ANOVA test followed by the Tukey’s Multiple Comparison Test; * *p* < 0.05, ** *p* < 0.01 were considered significant (NHDF vs. patients). Pt1: patient 1; Pt2: patient 2; Pt3: patient 3; NHDF: normal human dermal fibroblast.

**Figure 4 ijms-23-12774-f004:**
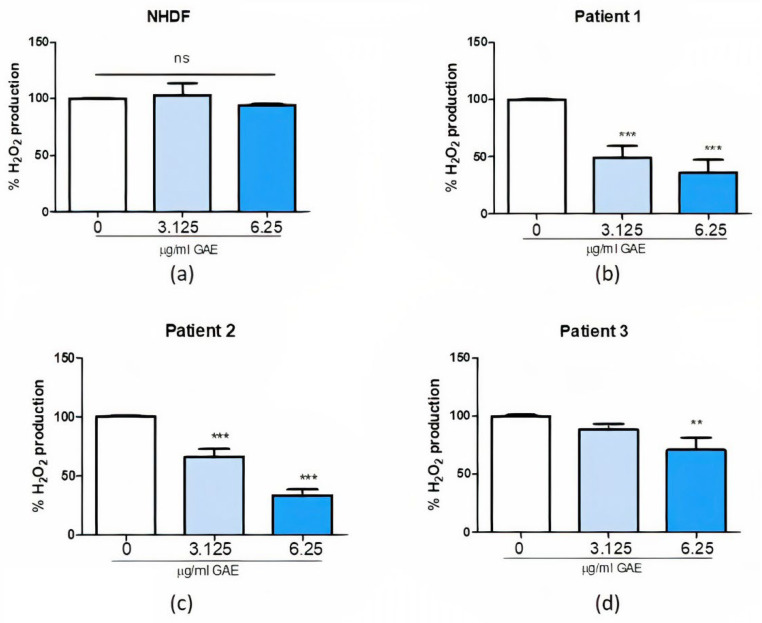
ROS production in NHDF (**a**) and patients (**b**–**d**) after phenolic sumac extract treatment. Production of intracellular ROS was calculated after 24 h treatment. All data represent the means of at least 3 replicates ± standard deviation. The analysis of variance was carried out using the One-way ANOVA test followed by the Tukey’s Multiple Comparison Test; ** *p* < 0.01, *** *p* < 0.001 were considered significant (untreated control vs. treated samples). NHDF: normal human dermal fibroblast; GAE: gallic acid equivalents.

**Figure 5 ijms-23-12774-f005:**
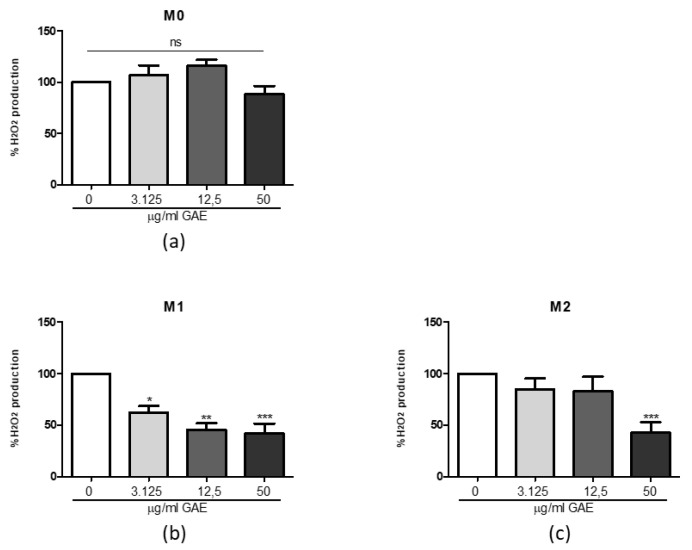
ROS production in M0 (**a**), M1 (**b**) and M2 (**c**) macrophage phenotypes after phenolic sumac extract treatment. Production of intracellular ROS was calculated after 24 h treatment. All data represent the means of at least 3 replicates ± standard deviation. The analysis of variance was carried out using the One-way ANOVA test followed by the Tukey’s Multiple Comparison Test; * *p* < 0.05, ** *p* < 0.01, *** *p* < 0.001 were considered significant (untreated control vs. treated samples). GAE: gallic acid equivalents.

**Figure 6 ijms-23-12774-f006:**
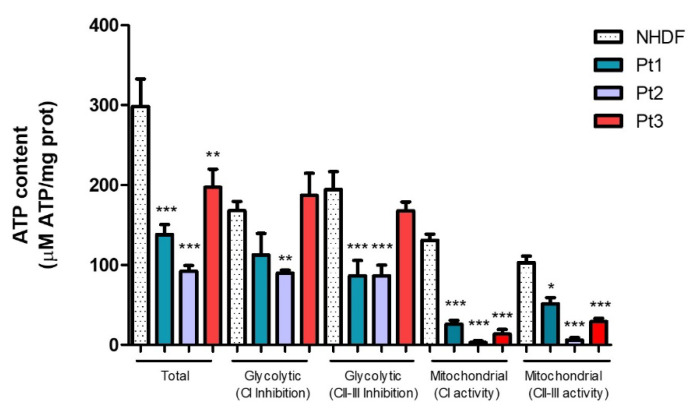
ATP production in NHDF and PD patients. Total, glycolytic and mitochondrial ATP levels, expressed as μM ATP/mg protein (prot), were assayed. All data represent the means of at least 3 replicates ± standard deviation. The analysis of variance was carried out using the One-way ANOVA test followed by the Tukey’s Multiple Comparison Test; * *p* < 0.05, ** *p* < 0.01, *** *p* < 0.001 were considered significant (patients vs. NHDF). Pt1: patient 1; Pt2: patient 2; Pt3: patient 3; NHDF: normal human dermal fibroblast; CI: Complex I; CII-III: Complex II-III.

**Figure 7 ijms-23-12774-f007:**
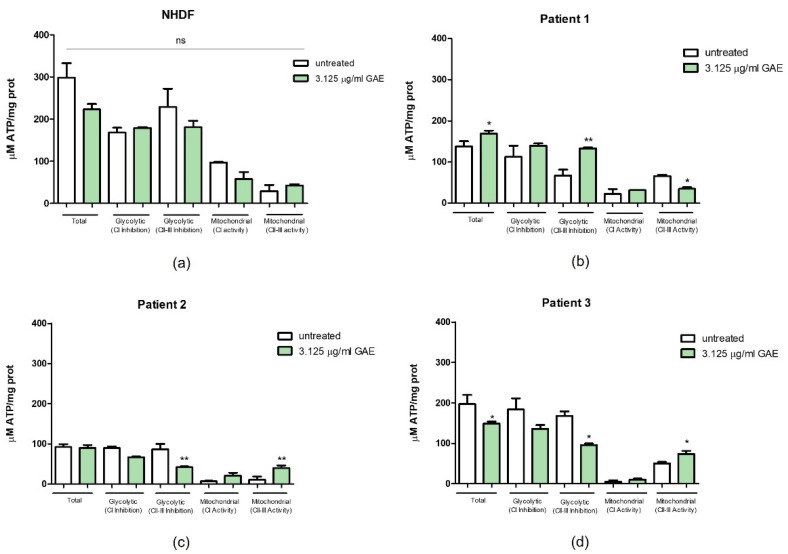
Effect of phenolic sumac extract on ATP production in NHDF (**a**) and PD patients (**b**–**d**). After 24 h treatment the production of total, glycolytic and mitochondrial ATP levels, expressed as µM ATP/mg protein (prot), was measured. All data represent the means of at least 3 replicates ± standard deviation. The analysis of variance was carried out using the One-way ANOVA test followed by the Tukey’s Multiple Comparison Test; *p* = ns (not significant); * *p* < 0.05, ** *p* < 0.01 were considered significant (untreated vs. treated samples). NHDF: normal human dermal fibroblast; GAE: gallic acid equivalent; CI: Complex I; CII-III: Complex II-III.

**Figure 8 ijms-23-12774-f008:**
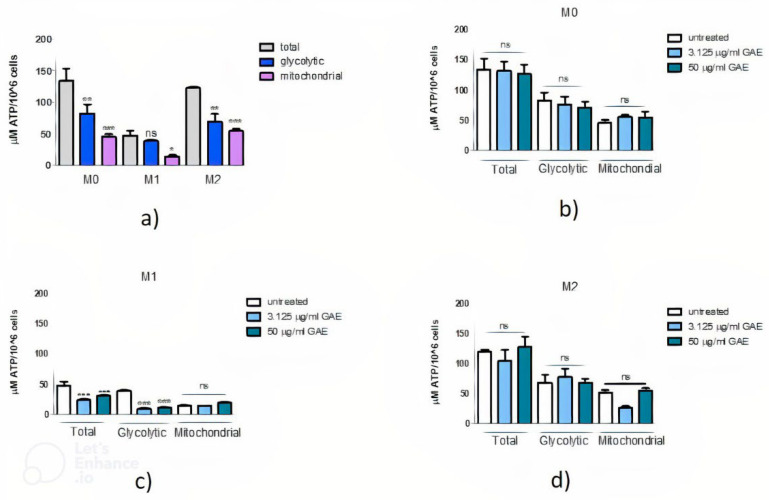
Effect of phenolic sumac extract on ATP production in M0, M1 and M2 macrophages. After 24 h treatment the production of total, glycolytic and mitochondrial ATP levels, expressed as μM ATP/10^6^ cells, was measured. Panel (**a**) shows ATP levels in M0, M1 and M2 macrophage phenotypes before the treatment with SE. ATP levels after treatment with SE are shown in M0 (**b**), M1 (**c**) and M2 (**d**). All data represent the means of at least 2 replicates ± standard deviation. The analysis of variance was carried out using the One-way ANOVA test followed by the Tukey’s Multiple Comparison Test; * *p* < 0.05, ** *p*< 0.01, *** *p*< 0.001 were considered significant (untreated vs. treated samples in different conditions). GAE: gallic acid equivalents.

**Table 1 ijms-23-12774-t001:** Antioxidant activity of sumac extract (SE) and gallic acid (GA). The antioxidant activity was evaluated with DPPH assay and TEAC assay and expressed for both as IC50 (mg/mL) and TE (mmol TE/g extract). Data are expressed as averages of three experiments ± standard deviation. SE and GA means are significantly different for *p* ≤ 0.05 (Student’s *t*-test analysis). IC50: half maximal inhibitory concentration; TE: Trolox Equivalents.

	DPPH Assay		TEAC Assay	
	TE (mmol TE/g)	IC50 (mg/mL)	TE (mmol TE/g)	IC50 (mg/mL)
**SE**	1.02 ± 0.05	0.41 ± 0.02	1.76 ± 0.10	0.21 ± 0.005
**GA**	26.68 ± 1.05	0.016 ± 0.001	29.19 ± 2.55	0.012 ± 0.0003

**Table 2 ijms-23-12774-t002:** Genotypic characterization and disease severity for the three patients enrolled in this study. Sex, age at PD onset, age at skin biopsy, following the Hoehn and Yahr (HY) and Unified Parkinson Disease Rating (UPDRS) Scales, are indicated. The control is a 53-year-old female. Pt1: patient 1; Pt2: patient 2; Pt3: patient 3; NP: not published.

Patient	Code	Sex	Age Onset (yr)	Age at Skin Biopsy (yr)	Disease Duration (yr)	PARK2 Mutation	H-Y Stage	UPDRS
**Pt1**	NP	F	28	36	8	**Cys253Tyr/ex5del**	1	5
**Pt2**	IT-048-NA099 [41,42]	M	47	64	17	**ex3-4del/ex3-4del**	3	30
**Pt3**	IT-031-NA025 [43]	M	33	65	32	**ex2del/ex2-4del**	4	35

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
