# Peer review of "Rhus Coriaria L. Extract: Antioxidant Effect and Modulation of Bioenergetic Capacity in Fibroblasts from Parkinson’s Disease Patients and THP-1 Macrophages"

_ijms, 2022, doi:10.3390/ijms232112774_

Round 1

Reviewer 1 Report

Camilla Isgrò tried to clarify the role of SE on oxidative stress and investigate its effect on THP-1 differentiated macrophages. These analyses had an exploratory aim and could eventually lead to further studies on primary macrophages obtained from PD patients. Some shortcomings are listed below.

1. This paper mainly focused on oxidative stress, but the detection indicators were only ROS and ATP, which are too few to fully get the conclusion.

2. The title of the article is mainly about antioxidant and metabolism, but the results of metabolism-related indexes were few.

3. The relationship between oxidative stress and polarization is not clear. The author did not explained the results between ROS/ATP and the change of polarization(M1/M2).

4. Current amount of results data is small. If the authors try to explain the effect on PD, it is suggested to add in vivo experiments.

Author Response

Responses for the Reviewer 1:

1)         “This paper mainly focused on oxidative stress, but the detection indicators were only ROS and ATP, which are too few to fully get the conclusion.”

Our work is focused not on the oxidative stress but on the antioxidant effect of sumac extract’s polyphenols, so we decided to evaluate intracellular ROS levels. Since ROS are mainly producted in mitochondria, we have also investigated if this antioxidant effect could have an impact on ATP levels, particularly in the mithocondrial quote (see revised Figure 6, 7, 8)

2)         “The title of the article is mainly about antioxidant and metabolism, but the results of metabolism-related indexes were few.”

Our study is a pilot study with the aim to investigate for the first time the antioxidant effect of sumac as potential nutraceutical approach to Parkinson’s Disease Through modulation of the cellular bioenergetics state (ATP levels). For the future, we want to extend our research on other parameters, such as evaluating mitochondrial functionality (i.e. Oxidative Phosphorylation Rate) and mitochondrial dynamics processes

3)         “The relationship between oxidative stress and polarization is not clear. The author did not explained the results between ROS/ATP and the change of polarization (M1/M2).”

This topic goes beyond the scope of our work, since our aim was not to demonstrate a change in polarization state of macrophages after SE treatment. We evaluated the possible changes in ROS and bioenergetic levels in this cell types, to proceed in the future with further analysis on PD patient’s derived macrophages (including not only PD genetic forms but also idiopathic ones), that would represent a sample easier to obtain than skin biopsy.

4)         “Current amount of results data is small. If the authors try to explain the effect on PD, it is suggested to add in vivo experiments.”

Thank you for the suggest, in this moment we have conducted a study as screening on the effect of sumac extract on Parkinson’s disease, since there is no evidence in literature about it. Confirming our results with other in vitro studies could represent the basis for future in vivo studies, that will include not only familial but also idiopathic forms of Parkinson’s Disease.

Reviewer 2 Report

The manuscript titled, Antioxidant and metabolic effects of Rhus coriaria L. extract in 2 THP-1 macrophages and fibroblasts from Parkinson's Disease 3 patients authors investigated the effect of sumac extract on THP-1 differentiated macrophages, which show different embryogenic origin to fibroblasts. The overall quality of work is acceptable for publication in IJMS. However, the following concern should be addressed during revision:

1.      What is sumac? Describe at the beginning.

2.      Introduction should be crisp.

3.      A detailed description of Flow cytometry methods should be provided.

4.      I found the minor issues, many typos and grammatical errors are seen in the paper. There are grammatical mistakes and typographical errors in the manuscript. The author should recheck this manuscript carefully and remove all such errors.

5.      Future directions and future implications should be described in a clear manner with a strong conclusion.

6.      A uniform presentation is required. The author should proofread the manuscript before the final submission. 

Author Response

Responses for the Reviewer 2:

1)         “What is sumac? Describe at the beginning.”

In the introduction paragraph (lines 103 to 123) we have described Rhus coriaria L. (Sumac), anyway we have enriched the introduction with other information (line 106-111)

2)         “Introduction should be crisp.”

Thank you for your suggestion, we have modified the introduction paragraph as indicated to make it crisper.

3)         “A detailed description of Flow cytometry methods should be provided.”

We have not described Flow cytometry since we have not used this method in our research. For the differentiation of macrophages from THP-1 monocytes, we have used protocols already standardized from our research group, as you can see in references 54.

4)         “I found the minor issues, many typos and grammatical errors are seen in the paper. There are grammatical mistakes and typographical errors in the manuscript. The author should recheck this manuscript carefully and remove all such errors.”

We have carefully checked the manuscript and modified grammatical and typographical errors.

5)         “Future directions and future implications should be described in a clear manner with a strong conclusion.”

As suggested by the reviewer, we have amplified the “conclusions paragraph” adding more information about the future directions.

6)         “A uniform presentation is required. The author should proofread the manuscript before the final submission.”

The manuscript has been proofread as required.

Line 3- title modified

Line 28- paragraph deleted and inserted at line 32

Line 34- paragraph changed

Lines 105-111: paragraph amplified

Line 129- paragraph modified

Line 132- revisioned english

Lines 147-148- paragraph modified

Line 251- word replacement

Lines 252-257: text modified

Line 259: graph replaced

Line 260: paragraph edited

Lines 268-270: text modified

Line 273: graph replaced

Lines 275-277: text modified

Lines 282-287: text modified

Line 292: text inserted

Line 292: text inserted

Line 293: text modified

Lines 302-303: text mofied

Line 308: text modified

Line 310: graph replaced

Line 313: text modified

Line 345: text deleted

Line 347: text inserted

Lines 381-391: text modified

Lines 405-409: text modified

Lines 411-414:text modified

Lines 416-422: text modified

Line 430: text inserted

Lines 551-558: text modified

Round 2

Reviewer 1 Report

1. Please add one or two more indicators in the results part besides ROS and ATP.

2. Please revised the title to mainly focus on antioxidant, since there were few results of metabolism-related indexes.

3. Please explain the reason why the authors detected the M1/M2 markers in the discussion part.

Author Response

  1. Thank you for the suggestion. As already explained in our previous reply, currently we have no other well consolidated data to include in this manuscript. Ours is a pilot study which aimed to investigate, for the first time, the effect of sumac extract in Parkinson’s disease patient’s cells, since there is no evidence about this in literature. However, further analyses are in progress, but it takes too long to validate other dosages in order to include those in this manuscript. We are willing to keep working for publish any promising new data in future publications.
  2. The title has been revised, as suggested by the reviewer.
  3. We guess the reviewer was suggesting we should explain in the discussion paragraph why M1/M2 markers have not been detected, so we added the explanation as required (see line 436-439).

Round 3

Reviewer 1 Report

The abbreviations in figures should be explained in figure legends (such as NHDF, GAE, SE,...). And the abbreviations should be shown in the first time mentioned with full-name, and only used abbreviations later.

Author Response

  1. “The abbreviations in figures should be explained in figure legends (such as NHDF, GAE, SE,...). And the abbreviations should be shown in the first time mentioned with full-name, and only used abbreviations later.”

We reported and explained in figure legends all the abbreviations included in graphs and figures. All the abbreviations used in the manuscript were first mentioned with full name in the main text.